# Clinical and Laboratory Features of Three Rare Chinese V210I gCJD Patients

**DOI:** 10.3390/pathogens9100800

**Published:** 2020-09-28

**Authors:** Kang Xiao, Wei Zhou, Li-Ping Gao, Yue-Zhang Wu, Yuan Wang, Cao Chen, Chen Gao, Qi Shi, Xiao-Ping Dong

**Affiliations:** 1State Key Laboratory for Infectious Disease Prevention and Control, , National Institute for Viral Disease Control and Prevention, Chinese Center for Disease Control and Prevention, Chang-Bai Rd 155, Beijing 102206, China; xiaokang@ivdc.chinacdc.cn (K.X.); zhouwei@ivdc.chinacdc.cn (W.Z.); gaoliping2014521@sohu.com (L.-P.G.); wuyuezhang5@foxmail.com (Y.-Z.W.); wangyuan@ivdc.chinacdc.cn (Y.W.); chencao@ivdc.chinacdc.cn (C.C.); gaochen@ivdc.chinacdc.cn (C.G.); 2Collaborative Innovation Center for Diagnosis and Treatment of Infectious Diseases, Zhejiang University, Hangzhou 310027, China; 3Center for Biosafety Mega-Science, Chinese Academy of Sciences, Wuhan 430071, China; 4China Academy of Chinese Medical Sciences, Dongzhimeinei, South Rd 16, Beijing 100700, China; 5Center for Global Public Health, Chinese Center for Disease Control and Prevention, Chang-Bai Rd 155, Beijing 102206, China

**Keywords:** genetic Creutzfeldt–Jacob disease, V210I, *PRNP*, prion

## Abstract

Genetic human prion diseases are a group of inherited encephalopathies directly associated with different mutations in PrP-encoding gene *PRNP*, including more than 50 different mutations worldwide. Some genotypes of mutations show ethno-correlation, and among them, genetic Creutzfeldt–Jacob disease (gCJD) with V210I mutation is frequent in European countries but rare in East Asia. Here, we comparatively analyzed the clinical and laboratory features of three Chinese patients with V210I mutant identified via the Chinese National CJD Surveillance System (CNS-CJD) in 2019. Two cases were Han Chinese and one was Hui Chinese, without blood kinship. The onset ages of three cases were 69, 64, and 59 years old, respectively. The clinical features of V210I gCJD were similar to sporadic CJD (sCJD), displaying typical clinical symptoms and signs, except that Case 3 did not show myoclonic movement. All three cases displayed sCJD-associated abnormalities on MRI and positive CSF 14-3-3, while two cases recorded typical EEG abnormalities. Only one case was positive in CSF real-time quaking-induced conversion (RT-QuIC). Appearances of mutism in three cases were relatively fast, with the intervals of 30 to 50 days after onset. Family history was not reported in all three cases. Those V210I gCJD cases are rare in China, and probably the first three in East Asia.

## 1. Introduction

Human prion diseases consist of sporadic, inherited and acquired forms, with most frequent subtype of sporadic Creutzfeldt–Jacob disease (sCJD). Inherited form accounts for about 5–15% of prion diseases, which are subtyped to genetic CJD (gCJD), Gerstmann–Straussler–Scheinker syndrome (GSS), and fatal familial insomnia (FFI) [1,2]. The clinical features and neuropathological abnormalities of different genetic prion diseases may vary largely depending on the various genotypes [1,2]. Although gCJD cases have been reported worldwide, some genotypes of mutations show an ethno-correlation [3,4,5,6,7]. V210I gCJD is one of the commonest subtypes in European and American countries [8,9,10,11,12]. It has been also identified in Japan and Korea [13,14,15], however it is never described in Chinese, although more than 200 genetic prion diseases consisting of 16 different mutations were diagnosed via the Chinese National Surveillance for CJD in the past 15 years [3,16]. In 2019, three Chinese V210I gCJD cases were identified and diagnosed with rapid clinical progression. Here, we reported the clinical, laboratory and genetic features of those V210I gCJD three patients.

## 2. Materials and Methods

### 2.1. Ethics Statement

Usage of the patients’ information in China CJD Surveillance System has been approved by the Research Ethics Committee of National Institute for Viral Disease Control and Prevention, China CDC (CCDC). The written informed content of each suspected case has been asked and signed by the family member or the relative of the patient according to the requirement of CJD surveillance.

### 2.2. Data Collection

As described previously [17,18], the clinical data of the suspected CJD patients were collected by the neurologists in the local hospitals and the epidemiological data were collected by the staff from local provincial CDCs. The final diagnosis was given by the expert team consisting of neurologists, epidemiologists and laboratory staffs based on the diagnostic criteria for CJD recommended by WHO and that issued by Chinese National Health Commission 2017. The follow-up surveys for the patients were performed by the staff of CNSNC via telephone.

### 2.3. Routine Laboratory Tests for CJD

The specimens of blood and CSF of the patient were collected by local medical staff. Western blot for CSF protein 14-3-3, PCR amplification for *PRNP* gene and further sequencing were conducted in the national reference laboratory for human prion disease in China CDC, according to the standard operation procedure (SOP) [18].

### 2.4. RT-QuIC Assays

RT-QuIC assay was performed according to the working procedures described previously [19]. Briefly, each reaction contained 10 µg of rHaPrP90-231, 1X PBS, 170 mM NaCl, 1 mM EDTA, 0.01 mM ThT, 0.001% SDS, together with 15 µL CSF samples in a final volume of 100 µl. The assay was conducted in a black 96-well, optical-bottomed plate (Nunc, 265301) on a BMG FLUOstar plate reader (BMG LABTECH, Offenburg, Germany). The working conditions were: temperature, 55 °C; vibration speed, 700 rpm; vibration/incubation time, 60/60 sec; total reaction time, 60 h. The ThT fluorescence value (excitation wavelength, 450 nm; emission wavelength, 480 nm) each reaction was automatically counted every 45 min and further presented as relative fluorescence units (rfu). Each sample was tested in quadruplicated simultaneously. The cutoff value was set as the mean value of the negative controls plus 10 times the standard deviation. A sample was considered to be positive when ≥2 wells revealed positive reaction curves. As the positive control, a dilution of 10^−5^ brain homogenate of the scrapie agent 263K-infected hamster was used, while 10^−5^ diluted brain homogenate of normal hamster was used as the negative control.

## 3. Results

Since 2006, national surveillance for CJD has been conducted, and more than 200 genetic human prion diseases have been identified (unpublished data). By the end of 2018, 16 different subtypes of genetic prion diseases involving in point-mutations and insertion of additional octapeptide repeat(s) (OR) were observed. In 2019, three Chinses V210I gCJD cases were detected with relatively rapid progression. All three cases were living in different provinces without detectable blood linkage (Table 1).

### 3.1. Clinical Data

Case 1 was a 69-year old male. He complained to have dizzy and tongue tip numbness particularly after overwork one month ago. He was recorded to display some neurological and mental abnormalities soon afterwards, such as cognitive problems, memory loss, speech disorder, salivation, wearing summer clothes in the wintertime, and needing to be reminded by others to swallow while eating and drinking. The patient denied having any infection before onset. During hospitalization, he appeared to display drowsiness, speech loss, and was uncooperative in physical examination. BP: 148/87 mmHg. High muscle tension in the extremities, grade 2 muscle strength in right lower limb, and enhanced tendon reflex of right limbs were recorded in neurological examinations. MRI recoded high signals in bilateral caudate head and lateral ventricular margin. EEG showed moderate and severe abnormality, including diffusive slow waves, sharp waves in left frontal lobe and front temper lobes, and untypical periodic sharp wave complexes (PSWCs). During hospitalization, progressive dementia, myoclonic movements in bilateral upper limbs, pyramidal and extrapyramidal symptoms, and cerebellum and visual disturbances were noticed. His symptoms were not improved, and he discharged from hospital couples of weeks later. A follow-up survey one month later reported akinetic mutism that the patient laid in bed with whole body rigidity. The patients died 75 days after onset.

Case 2 was a 64-year old male. Forty days ago, he appeared to have blurred vision, visual hallucination, sleep problems, and was walking unstable without obvious incentive. Head MRI (DWI) examination reported intracranial abnormal signals. About 10 days later, he displayed more severe mental problems that he could communicate normally during daytime but raved and threw things in the night. He received therapeutics for cerebrovascular disease and for autoimmune cerebritis without any improvement. Gradually, he appeared to display apathy, speech loss, drowsiness, and dysphagia. Neurological examination revealed gazing to the right, occasional voluntary limb movement, hypermyotonia, but weak tendon reflex in bilateral upper limbs, positive in Babinski’s and Chaddock’s signs. Other physical examination was incomplete because of uncooperative behaviour. An MRI scan reported high signals at the surfaces of bilateral parietal lobes and frontal lobes. EEG recorded extensively typical PSWCs. During hospitalization, progressive dementia, myoclonic movement, pyramidal and extrapyramidal symptoms, cerebellum, and visual disturbances were noticed. His general conditions aggravated rapidly that akinetic mutism was obvious later. A follow-up survey 10 months after onset recorded a severe mutism condition with total losses of cognitive, speech, and movement capacities. 

Case 3 was 59-year male. About 4 weeks ago, he appeared to display side to side body shaking in seating and standing positions without a known reason, accompanying occasional dizziness. The symptoms aggravated rapidly that he displayed walk unstable, increased step spacing and clumsy speech. Therapeutics for cerebral infarction and for autoimmune cerebritis did not reveal any improvement. Neurological examination found unrestricted eye movement, binocular horizontal nystagmus, unstable in bilateral heel-knee-tibia test, positive in Romberg sign. MRI scan reported high signals in parts of parietal and frontal lobes, as well as in bilateral caudate. EEG recorded diffusive slow waves and typical PSWCs. During hospitalization, progressive dementia, pyramidal and extrapyramidal symptoms, namely cerebellum and visual disturbances, were reported. A follow-up survey three months after onset reported the patient appeared akinetic mutism. The patient died 9 months after onset.

### 3.2. Laboratory Tests

Lumber punctures were performed for all three patients. Cerebrospinal fluid (CSF) biochemistry assays did not show any abnormality as the pressure, the numbers of white and red blood cells, and the concentrations of protein and glucose were all in the normal ranges. Further, a 15-μL CSF sample from each patient was subjected into 14-3-3 specific Western blots. This revealed positive bands in the CSF samples of all three cases (Figure 1). Further, aliquot of 15 μL CSF sample was employed into RT-QuIC assays using recombinant truncated hamster PrP from aa 90 to 231 (HaPrP90-231) as the substrate. As shown in Figure 2, Case 2 showed the positive reaction in RT-QuIC, whereas Cases 1 and 3 were negative. The average positive conversion time from four duplication wells was 12.75 h and the peak was 120,000 rfu.

### 3.3. Family History

Individual family histories of three cases were carefully investigated. The parents of Case 1 died without fixed diagnosis at the age of 70 years old. Three brothers and two children were healthy. All the three cases denied family history of similar neurological disease.

### 3.4. PRNP Sequencing 

The blood samples of three patients were taken during hospitalization. The PCR products amplified with the specific primers for *PRNP* gene were subjected into sequencing. The same missense mutation at codon 210 (GTT to ATT) were identified in the blood samples of those patients, leading to a substitution from valine (Val) to isoleucine (Ile) (Figure 3). All three patients showed the same the polymorphism of codon 129 (Met/Met) and of codon 219 (Glu/Glu). No other nucleotide exchange was identified in the remaining *PRNP* sequences. The family members of three cases refused to donate blood samples for *PRNP* sequencing.

## 4. Discussion

In this study, we have reported the first three Chinese V210I gCJD identified in 2019. All three cases were male. According to prospective surveillance in Germany, the ratio of female and male was 1:1.1 [9]. It seemed that V210I did not have a gender tendency. In the Chinese surveillance system, the reported number of cases was limited. Therefore, a conclusion cannot be made yet. However, considering the location of the *PRNP* gene, we believed that there was no gender difference for V210I mutation. All three cases show acute onset and rapid progression in the first two months. Similar as other reported cases [13,20], The clinical features of three Chinese V210I gCJD are more like sCJD, with rapid progressive dementia and four other main neurological symptoms. Mental problems and ataxia are frequent initial disorders. More importantly, the interval from onset to having evident mutism is fairly short (less than 50 days), which is, from our perspective, shorter than that of the majority of Chinese sCJD patients [21]. Based on our surveillance data [17,18], the appearance of mutism is observed in about one third of Chinese patients with sCJD (definite and probable diagnoses). Obviously and fast occurring mutism in V210I gCJD cases reflect a rapid progression and aggravation clinically. Rapid progressive neurological signs have been also documented in V210I gCJD cases from other countries [13,20]. In addition, other clinical features of three Chinese V210I gCJD cases are also similar as the patients reported from other countries, such as the onset age (around 60 years old), short duration, and mostly lacking relevant family history [9]. 

Examinations of EEG and MRI of three Chinese V210I gCJD cases also display a sCJD-like patterns, which are documented shortly after onset. A study containing 29 sCJD cases and 29 cases of E200K and V210I gCJD has proposed similarity in MRI abnormality [22]. Our recent study of 30 Chinese E200K gCJD patients also demonstrated very high positive ratios (86.7%) of sCJD-associated MRI changes [23]. CSF 14-3-3 positive is observed in all three cases here and the positive rate seems to be higher than that of Chinese sCJD [16] and E200K gCJD [23]. High ratios of positive CSF 14-3-3 and PSWCs in EEG of V210I gCJD, but relatively lower ratio of MRI abnormality, have been reported in an early study [10], probably because of the updating of MRI diagnostic criteria. Only one Chinese case in this study shows positive in CSF RT-QuIC. Certainly, the numbers of Chinese V210I gCJD cases is too few to obtain a final conclusion. 

Previous studies have already revealed that the V210I mutant is the most commonly reported one in Italy, and also reported in other European countries, such as Germany, France, Austria and Switzerland [8,9,10], but rare in Eastern Asia. A review has shown a statistical significance in the distribution of V210I gCJD between East Asian and European countries [6]. Those data highlight again an ethno-correlation in the distribution of V210I mutant. In fact, the diversities of the polymorphism and disease-associated mutants within *PRNP* is not only observed between Caucasians and East Asians, but also among the East Asians, e.g., T188K mutant is the most frequent gCJD mutation in Chinese [3,4], but rarely identified in Japanese and Korean patients, whereas P105L and M232R are commonly detectable in Japanese [14,24,25,26] but extremely rare in Chinese patients. 

We have no neuropathological and pathogenic data of V210I gCJD in this study. Some postmortem assays of V210I gCJD demonstrate sCJD-like neuropathological patterns, such as spongiform degeneration of neurons and reactive astrocytosis, diffuse positive and plaque-type PrP^Sc^ deposit, and type-1 PrP^Sc^ patterns [10,13]. Although V210I mutant is estimated to be low penetrance mutant in the Italian population (only approximately 10%) [20,27], an early study has confirmed that inoculation of the brain homogenates from V210I gCJD patients onto transgenic (Tg) mice expressing a chimeric human-mouse PrP sufficiently induced transmission whose PrP^Sc^ deposit patterns in brains are more like that of sCJD but different from those of E200K gCJD and D178N FFI [28]. Further structural assays have proposed that V210I mutant induces reorientations of several residues in the hydrophobic core that influences α(2)-α(3) inter-helical interactions, and involves alteration of conformation of the β(2)-α(2) loop region and the subsequent exposure of the hydrophobic cluster to solvent, which facilitates the spontaneous generation of PrP^Sc^, particularly under acidic pH conditions in endosomal compartments [29,30]. However, the linkage between structural changes in vitro and pathogenesis in vivo of V210I mutant still needs to be explored.

## Figures and Tables

**Figure 1 pathogens-09-00800-f001:**
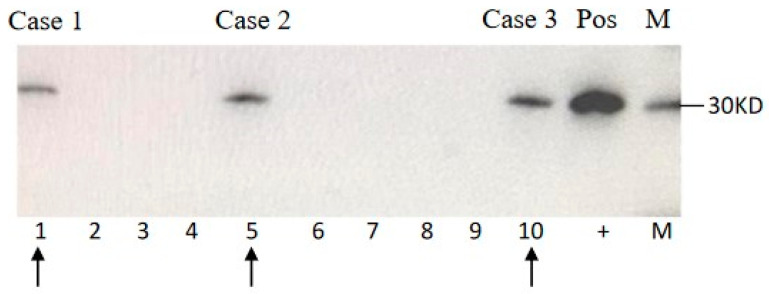
Western blot for 14-3-3. 15 μL of CSF sample each patient was subjected into 12% SDS-PAGE. Blots were incubated in 1:1,000 diluted 14-3-3 polyclonal antibodies (Santa Cruz, CA, USA) and further incubated in 1:5,000 diluted HRP-conjugated goat anti-Rabbit IgG (PerkinElmer, Germany). Immunoreactive bands were visualized by ECL method (PerkinElmer, Germany). M: molecular standards. (+): positive control of 10% goat brain homogenate. Arrows indicate the CSF samples of the individual V210I gCJD patients.

**Figure 2 pathogens-09-00800-f002:**
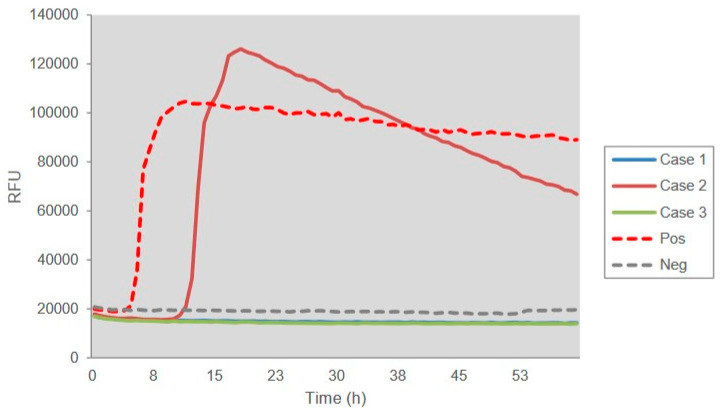
Results of RT-QuIC assays of CSF samples of three V210I gCJD Chinese patients. 10^−5^ diluted brain homogenate of scrapie agent 263K infected hamster was used as positive control and that of normal hamster was used as negative control. ThT value is showed in Y-axis and hour post-reaction is indicated in X-axis.

**Figure 3 pathogens-09-00800-f003:**
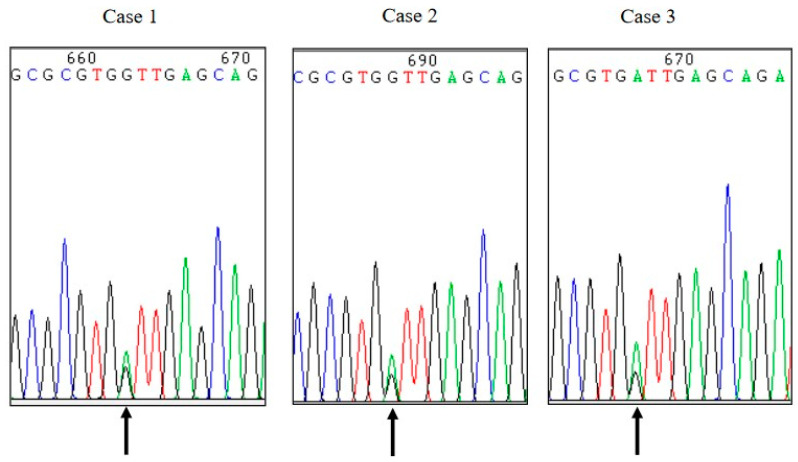
Graphic presentations of the sequences of *PRNP* from 3 Chinese V210I gCJD patients by direct sequencing. G/A heterozygote at codon 210 are detected, leading to an exchange from Val (V) to Ile (I). The arrows below the curves indicate the positions where both G and A are co-present.

**Table 1 pathogens-09-00800-t001:** The main features of three Chinese V210I gCJD patients.

	Gender, on Set Age	Dementia ^1^	Other Major CJD-Associated Problems ^2^				EEG	CSF		MRI		Polymorphism		Duration	
			I	II	III	VI	PSWCs	14-3-3	RT-QuIC	Ribbon-like signal	High signals in caudate/putamen	Codon 129	Codon 219	Interval from onset to mutism	Situation
Case 1	M, 69 y	+	+	+	+	+	untypical	+	-	+	+	M/M	E/E	app. 30 days	Died 75 days after onset
Case 2	M, 64 y	+	+	+	+	+	typical	+	+	+	-	M/M	E/E	app. 50 days	Alive with mutism
Case 3	M, 59 y	+	-	+	+	+	typical	+	-	+	+	M/M	E/E	app. 40 days	Died 9 months after onset

^1^ Rapid progressive dementia. ^2^ I: myoclonic movement; II: Cerebellum and visual disturbances; III: Pyramidal or extramidal disfunction; VI: Akinetic mutism.

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
