# Peer review of "Clinical and Laboratory Features of Three Rare Chinese V210I gCJD Patients"

_pathogens, 2020, doi:10.3390/pathogens9100800_

Round 1
Reviewer 1 Report
The article is well designed, performed and executed. Overall, the study will be of high interest to the field of neurogenic disorders related with human prion diseases. However, I want the authors address the following comments.
Minor comments:
Comment 1: The three cases described correspond to adult men, it could be that this disease has a sex-specific genetic or condition? The authors could include this point in the discussion.
Comment 2: Since the authors found plaque-type-PrPSC deposit in postmortem samples, similar to Alzheimer's disease (AD) [1, 2]. Could be interesting that the authors to comment on whether they found a history of depression or post-traumatic stress in the patients. Clinic history that could be related to the onset of the disease, as occurs in AD. It will be nice that the author can address this matter in discussion.
Comment 3: The authors comment in the discussion that there are previous studies that reveal that more common mutants are reported in some European countries, but rare in Eastern Asia. It could be interesting for researchers and doctors present in other continents to include globalized information from America, Africa and Australia.
[1] Frost, B.; Diamond, M.I. Prion-like mechanisms in neurodegenerative diseases. Nat. Rev. Neurosci. 2010, 11, 155–159.
[2] Duyckaerts, C.; Clavaguera, F.; Potier, M.C. The prion-like propagation hypothesis in Alzheimer’s and Parkinson’s disease. Curr. Opin. Neurol. 2019, 32, 266–271.
Author Response
Comment 1: The three cases described correspond to adult men, it could be that this disease has a sex-specific genetic or condition? The authors could include this point in the discussion.
Answer: Please see line 164-169.
Comment 2: Since the authors found plaque-type-PrPSC deposit in postmortem samples, similar to Alzheimer's disease (AD) [1, 2]. Could be interesting that the authors to comment on whether they found a history of depression or post-traumatic stress in the patients. Clinic history that could be related to the onset of the disease, as occurs in AD. It will be nice that the author can address this matter in discussion.
Answer: In the data of these three cases and their follow-up, we did not get the information about mental stress or depression. So we do not know whether depression or stress is related to the occurrence of V210I CJD.
Comment 3: The authors comment in the discussion that there are previous studies that reveal that more common mutants are reported in some European countries, but rare in Eastern Asia. It could be interesting for researchers and doctors present in other continents to include globalized information from America, Africa and Australia.
Answer:There are studies about V210I cases in some European countries such as Germany and Italy. However, other countries seldom reported V210I cases. Besides east-Asia, a case from North Africa was also reported in 1999. Therefore, researches of V210I in other continents will be valuable.
Reviewer 2 Report
This paper presents case reports of the clinical features and a reasonable amount of laboratory analysis of a novel genetic prion disease mutation within the Chinese population. Following clinical identification, the three cases were identified by sequencing for the G-A transition at position 210 of PRNP, plus CSF and RT-QuIC prion seeding analyses. While post-mortem analysis is mention in the comments, presence and distribution of spongiform change, prion deposition and glial responses would have been a useful addition to this paper for confirmation of prion disease and description of the histopathological features. Furthermore biochemical typing e.g. Western blotting for PK resistant PrP and description of its MW and glycoform profile also would have been a beneficial addition. As an initial case report this paper present evidence for the presence of V210I gCJD within the Chinese population. The reviewer recomends accepting this article after extensive revision for english language and grammar. Also clarification on the ethnic background of these cases as they are reported in the abstract as 2 Han-Chinese, 1 Hui-Chinese but in the Results (Line 83) as all three cases were Han-Chinese?
For clarification and cross comparison perhaps the authors could summarise the major findings and results from all 3 cases within a table and possibly compare these to sCJD or other gCJD types as mentioned throughout the manuscript?
Author Response
- It is a pity that none of these three cases had post-mortem, so we have no neuropathological and pathogenic data of V210I gCJD in this study.
- As for the ethnic background, two of the cases were Han-Chinese, and the other was Hui-Chinese. We have corrected the information in line 83.
- A table was inserted in line 83 and 289-291.
- We have polished the manuscript.